# How migration shapes modern contraceptive use among urban young women: Evidence from six African countries

**Jessie Pinchoff**[1]*, **Isabel Pike**[2], **Karen Austrian**[3], **Kathryn Grace**[4], **Caroline Kabiru**[5]

**1** Population Council, New York, NY, United States of America, **2** McGill University, Montreal, Canada, **3** Population Council-Kenya, Nairobi, Kenya, **4** University of Minnesota, Minneapolis, MN, United States of America, **5** African Population and Health Research Center, Nairobi, Kenya

* jpinchoff@gmail.com

## Abstract

### Background

Internal migration is an important part of the transition to adulthood for many young people in sub-Saharan Africa. This study examines how migration, in relation to marriage and parenthood, impacts modern contraceptive use and health facility visits amongst young urban women.

### Methods

We draw on Performance Monitoring for Action (PMA) surveys conducted in Burkina Faso, Côte d'Ivoire, Democratic Republic of Congo, Kenya, Nigeria, and Uganda (2019–2022). Our analysis is unique in being able to adjust for whether women wanted to get pregnant soon. Our sample includes women ages 15–24 years currently residing in urban areas (n = 6,225). We conducted logistic regression models clustered by village level identifier to explore the sequence of life events and the timing of migration in relation to current modern contraceptive use and recent health facility visit, a proxy for engagement with formal health services.

### Results

The timing of migration matters more than the sequence of these life events. Young urban women who experienced both migration and a birth, regardless of the order, had increased contraceptive use and recent health facility visit, compared to women who had only experienced one event or neither. Young women who migrated in the past year had 24% lower odds of using a modern method (Odds Ratio = 0.76; 95% confidence interval 0.63, 0.91), adjusting for demographic factors and adjusting for fertility preference (Wanting to get pregnant soon). Having had a birth was highly significant for health facility visit and among women who had had a birth, those who migrated in the last year had lower odds of a recent visit (OR = 0.68, 95% CI 0.41, 0.89). Results suggest an initially disruptive effect of migration.

**Data Availability Statement:** The data is available from the PMA website and can be accessed here: https://pma.ipums.org/pma/ We did not collect any primary data.

**Funding:** The author(s) received no specific funding for this work.

**Competing interests:** The authors have declared that no competing interests exist.

## Discussion

Our results suggest young women who recently migrated to urban areas may need additional support in accessing contraception and formal health services for themselves or their children.

## Background

Migrants who move from rural to urban areas in sub-Saharan Africa are often adolescents or young adults. This age pattern of migration is in part a reflection of the region's youthful population, but also the fact that early adulthood is dense with life events, related to education, work, and family, which can generate reasons to move [1–5]. For many, these life events mark important shifts in relationships—greater independence from parents, for example, and increased responsibilities towards children—and thus confer a level of socially recognized adult status. As a result, scholars have termed this period of life "the transition to adulthood," paying close attention to the timing and sequencing of events such as leaving school, starting paid work, and marrying with the understanding that they impact young people's current well-being and their future life trajectories [5, 6].

Much of the research on migration focuses on international migration, even though most migration is in fact internal or domestic. This focus in part reflects data constraints: internal moves are less likely to be recorded than international ones—in sub-Saharan Africa, but elsewhere too—and data disaggregated by age and sex is even more sparse [7, 8]. In part due to these data constraints, despite how common it is for young people to migrate in sub-Saharan Africa, the literature on how internal migration impacts the transition to adulthood is still relatively nascent. Drawing on life course theory, this small but growing body of research has begun to explore not only young people's reasons for migrating [3], but also increasingly how migration shapes the experience of other life events, including around sexual initiation, parenthood, work, and schooling [1, 2, 9, 10]. In parallel, there is a relatively large body of scholarship on how migration affects contraceptive use and healthcare access in sub-Saharan Africa, but it tends to focus on women of childbearing age broadly, rather than on young women [11–13]. This paper thus builds on these two areas of research by examining how internal migration, and specifically its timing in relation to marriage and birth, impacts the uptake of modern contraceptive methods and engagement with formal healthcare services amongst young women (15–24 years) in urban areas.

Our focus on modern contraceptive use in early adulthood reflects our understanding that young women's ability to control if and when to have children is crucial for their ability to navigate early adulthood in a way that expands their agency and reflects empowerment [14–16]. The second outcome of interest is recent visit to a health facility, which is an important measure of healthcare accessibility and uptake. In our data, recent health facility visits are not specifically tied to family planning, the visit can be for a woman's own health or for her child's health, indicating overall a connection and integration into formal health systems. For women with children, health facility visits are critical to access services including nutrition and immunization [17, 18]. Additionally, our analyses include whether women want a birth soon to better understand how the relationship between migration, contraceptive use and healthcare utilization might be mediated by differential fertility desires. Our study also takes advantage of a unique dataset, the Performance Monitoring for Action (PMA) surveys that include a measure of migration, information rarely collected in nationally representative surveys.

## Migration and the transition to adulthood

In a wide range of contexts, most people who migrate do so in early adulthood, a period when they are already experiencing other substantial changes, both physically and cognitively as well as in terms of their relationships and identities [3, 19–21].There are a variety of proposed reasons for this age pattern to migration, including that young people are especially likely to be looking for work opportunities and also that they are less likely to have established themselves in a particular location [5]. Examining migration from a life course perspective directs attention to when migration occurs, but also how it fits into a sequence of other life events [22]. A life course framework posits that these life events reflect individual agency, responding to the bounds of specific historical and cultural contexts [6], and also that the experience of these events have lifelong consequences, resulting in "cumulating advantages and disadvantages" [23]. Studying the implications of migration for adolescents and youth is particularly crucial in low- and middle-income countries given that cities in these countries are home to 90 percent of the world's 1.8 billion people ages 9–24 years [24].

Building on life course theory, scholars are increasingly exploring how the experience of different life events impact the decision to migrate and vice versa [2, 3, 23, 25]. This research on the connections between migration and other life events paints a complex picture, reflecting a variety of motivations and experiences: for example, rural young people move to urban areas for education opportunities [3], but equally, moving to urban areas is also at times motivated by seeking paid work and thus, associated with dropping out of school [2]. Within this context, scholars of youth experience in sub-Saharan Africa have begun to explore the gendered dimensions of migration [1–3, 20]. Female migration rates are growing faster than male migration rates, leading some researchers to describe "a feminization of migration" [25, 26]. Further, data suggest that women are increasingly migrating to take advantage of educational opportunities and better employment prospects, though marriage and other family reasons still remain an important driver of migration [1, 8, 20, 25, 27].

Overall, increasing contraceptive choice and access are critical for reproductive justice and rights of all women. Existing research has shown that pregnancies at younger ages are more likely to be unplanned and to come with health risks, including unsafe abortions [28]. Delaying the entry into parenthood also has significant implications for other aspects of life, with girls reporting that delaying childbearing would allow them to complete their education [29] and also to become financially independent [30–32]. Given these consequences of unintended pregnancy in early adulthood, scholars and policymakers have stressed the importance of strengthening access to modern contraception for adolescents and youth, alongside other interventions, including facilitating school re-entry [29, 33, 34]. However, these policies must also reflect the preferences of young women. The decision not to use contraception can also reflect reproductive agency, if women do want to become pregnant [35]. Addressing the contraceptive needs of young women including young migrant women is critical but must align with their needs and preferences.

## Migration as a risk factor?

The literature often paints a negative picture of migration and highlights the potential for migration to serve as a factor associated with riskier sexual behavior and unplanned pregnancy, and earlier age at first sex [10, 22, 36]. Male migration was believed to be one of the driving forces of the HIV epidemic across Africa and tied to increased rates of alcohol use or risky sexual behaviors [36–38]. Some studies of adolescent girls migration and SRH highlight the risks faced in conflict- or disaster-affected populations, wherein girls face much higher risk of STIs, HIV, and lower use of contraception [39].

It is not possible to categorize migrants wholly as a disadvantaged group as migration can both be driven by hardship, including poverty and climate shocks, but at the same time, facilitated by advantage, particularly wealth and education [3]. Indeed, the migrant women in our sample tend to be in wealthier quintiles than non-migrant women. Despite this mixed profile, there remain important challenges for young migrant women hoping to avoid a pregnancy. Some studies show that providers are often less keen to offer contraception to younger clients, often if they have not yet had a child [40–44]. Young women also express providers' attitudes as a concern [29]. Programs aimed at equipping young people with reproductive health knowledge often miss migrants if they do not speak local languages or if they are not integrated into formal institutions such as schools of health facilities where programs may be offered [45].

There is debate in the literature about the extent to which moving to an urban area offers a health advantage. People living in urban areas are generally considered to have better access to health services, but there is also the potential for growing disparities within urban areas, including through the concentration of poverty and unemployment in slum areas with low access to health services [46, 47]. Notably, while modern contraceptive use tends to be higher in urban areas in sub-Saharan Africa, access to family planning has been found to be considerably worse in slums [48]. If migrant women move to one of these areas, they might have worse access than they did in rural areas. However, migrant women can also have higher levels of contraceptive use and integration into healthcare systems than their urban non-migrant peers. One study from Ethiopia, for example, found that women who had moved from rural to urban areas were *less* likely to experience unmet need for family planning than non-migrant urban women; a finding that the authors attribute to rural-urban migrants higher "human capital" in terms of wealth and education [12].

While other studies have compared contraceptive use between rural-urban migrants and both rural and urban non-migrants [12, 49, 50], we focus our attention on urban areas to better understand the context that young rural-urban migrant women are navigating.

## Migration and contraception

Drawing on research on migration and fertility more broadly [13, 51], scholars have developed four main theoretical frameworks to conceptualize how migration might impact contraceptive use: socialization, adaptation, disruption, and selection [12, 52]. The socialization theory posits that attitudes and values that drive fertility behaviors are primarily formed while growing up and that the new economic and cultural environments that an individual encounters through migration do relatively little to shift them. In contrast, the adaptation theory suggests that these underlying attitudes and values are flexible, responding to the norms and demands of the destination. The disruption theory focuses more on the lived experience of migration: how migration might impact, for example, how often an individual has sexual intercourse due to separation from their spouse or how migrant women might breastfeed for a shorter period due to separation from their children, potentially changing their desire to use contraception [13]. Selection calls attention to the characteristics of women who choose to migrate and the fact that the same factors that are associated with the decision to migrate, such as higher education levels, are also often associated with women's use of contraception.

Scholars have emphasized that these theories are not mutually exclusive and that it is also often quite difficult to categorize an associated as rooted in one mechanism or another [13, 52]. Still, they remain useful broad schools of thought to conceptualize how migration might impact contraceptive use. Additionally, despite their different emphases, all of these theories point to the value of considering how long migrants have been in the new destination. For example, the adaptation theory suggests that migrants' contraceptive use might more closely

resemble those of their receiving communities rather than those they left behind as they adjust to their new environment. In terms of disruption, the neighborhoods that migrants live in at first might be less well serviced in terms of healthcare, but, over time, migrants might move to better serviced areas [53]. In our analyses, we integrate this attention to time by differentiating between migrants who have recently arrived and those who have been in the urban areas for longer, using one year as a cut-off point.

Perhaps indicating the value of all these different theories, empirical research has found no consistent association between migration and contraceptive use. Amongst the studies that have found a positive association, it often appears to be explained by selection. For example, one study in Turkey found that women who migrated within the country were more likely to use contraception, which the authors attributed largely to the fact that marriage is both a major reason for migration and associated with giving birth and thus using contraception [52]. Alongside this selection effect, the study also found support for women adapting to the new context as even after controlling for marriage and having a birth, migrant women were more likely to use modern contraception. Studies from Myanmar and Ethiopia have had similar findings, noting the difficulty in assessing whether this positive association is due to selection, adaptation, or access given their use of cross-sectional data [12, 54].

Studies that have found a negative association between migration and contraceptive use often find support for both the disruption and adaptation theories. For example, a study focused on Mozambique found that recent urban migrants were less likely to use modern contraception than urban non-migrant women, but that migrant women who had been in urban areas for longer were just as likely to use modern contraception as urban non-migrant women [55]. Further supporting the disruption explanation, the study found that living in a neighborhood with limited transport was associated with a reduced likelihood of using modern contraception. Similarly, a study from Guatemala found that women who migrated to a metro area were less likely to use modern contraception, but that duration of stay in the metro area was positively associated with using modern contraception [53]. Our paper builds on this work by focusing on young women and studying migration in relation to the experience of marriage and having a child.

## Migration and access to healthcare

Most literature on migration and how it relates to access to formal health services is focused on the global north and disparities in healthcare access for migrants from other countries. South Africa has perhaps the most extensive literature on delivery of health services to migrants, highlighting the potential disruption in continuity of care for migrants particularly if they face language barriers, cannot access information, or have negative interactions with healthcare providers, all of which may discourage health seeking–this makes migrants appear less likely to engage with the health system [56]. Another study found young female migrants faced access and availability of SRH services was limited due to distances, costs, and stigma [39]. Women with young children should have more engagement with health services after giving birth, to receive nutritional support, immunization, and treatments for leading causes of child death such as diarrhea or malaria [57–59]. Despite the urban advantage in density of healthcare providers, these are not evenly distributed or available.

Our study has two aims. First, to understand how recent migration, within the last year, might affect modern contraceptive use and recent health facility visit. We hypothesize that recent migrants may face more barriers to use of modern contraception and to recent access of health facilities, and that after this initial disruption the effect may fade given the improved access to resources and services available in urban areas. Second, we explore the sequence of

key events including migration, marriage, and first birth in relation to use of modern contraception and recent health facility visit. In theory, women who migrate or have a birth first may face different challenges accessing contraception or formal health services. Potentially, women who migrated then gave birth may be less likely to have recently visited a health facility, if they face barriers to access. On the other hand, women who migrate rural to urban may have more access to facilities because the infrastructure and density of services is greater [60]. If women who migrate to urban areas face a disruption in their ability to access health services, for any reason, this may affect their health.

## Methods

To examine the association between migration, in relation to birth and marriage, and the uptake of a modern contraceptive method, we draw on cross-sectional data from the Performance Monitoring for Action (PMA) surveys. PMA surveys are unique in their ability to explore migration and various characteristics cross-nationally in Africa, using most of the same questions, and capturing internal migration as well as sexual and reproductive health outcomes. In addition, though the PMA data are cross-sectional, they include questions on the ages of migration, marriage, and parenthood that allow us to construct a life event sequence variable, as detailed further below. As a result, the PMA surveys allow for an in-depth exploration of migration and young women's health across multiple countries. The data were collected from 2019 to 2022 across the six countries: Burkina Faso (2019, 2021 and 2022), Cote d'Ivoire (2020, 2021) Democratic Republic of Congo (Kinshasa and Kongo central region, 2019, 2020, 2021), Kenya (2019), Nigeria (Kanos and Lagos, 2019, 2020, 2021), and Uganda (2020 and 2021). If women had a repeat observation, we only kept the first one. No IRB approval was sought given that this is a secondary analysis using publicly available, de-identified data.

In the full sample, 69% of women ages 15–24 years had migrated. Around 60 percent of the young women (ages 15–24 years) surveyed had ever had sex with the proportion higher amongst those who had migrated compared to those who had not (63 percent compared to 56 percent). The distribution for age at birth, marriage, and migration shows that birth and marriage tend to occur around the same time, several years after sexual debut (**Fig 1**). Compared to the other life events, age at migration has less of a defined peak, rising steadily from 13 to 18 years and remaining at relatively high levels until 23 years. Migration is the most common of the three life events: By age 24, around 60 percent of respondents had migrated while around 30 percent have married and had a birth. We explored one, two and five year cut offs to highlight recent migration. Generally less than five years is considered recent, one paper on migration used less than 1 year, 1–5 years, and over 6 years as the categories [53] while another used less than 3 years, 3–5 and over 5 years [55]. While we tested three categories and various cut offs, we use migration within the previous year, to explore recent migration status and avoid potential recall bias for exact time of migration.

We restricted observations to women ages 15–24 years, who had ever had sex, were not pregnant, and currently residing in urban areas (n = 6,225). We restricted the analysis to respondents who have ever had sex because of our specific focus on birth in relation to migration. Furthermore, there were close to none who were using contraception and were not sexually active.

The two outcome variables are 1) current use of a modern contraceptive method and 2) recent health facility visits. Scholars and policy makers have justified a focus on modern methods because they are more effective in preventing pregnancy, but also because they pose greater challenges of access, in part because acquiring these methods generally requires money and/or medical assistance, unlike methods such as the rhythm method or withdrawal [61].

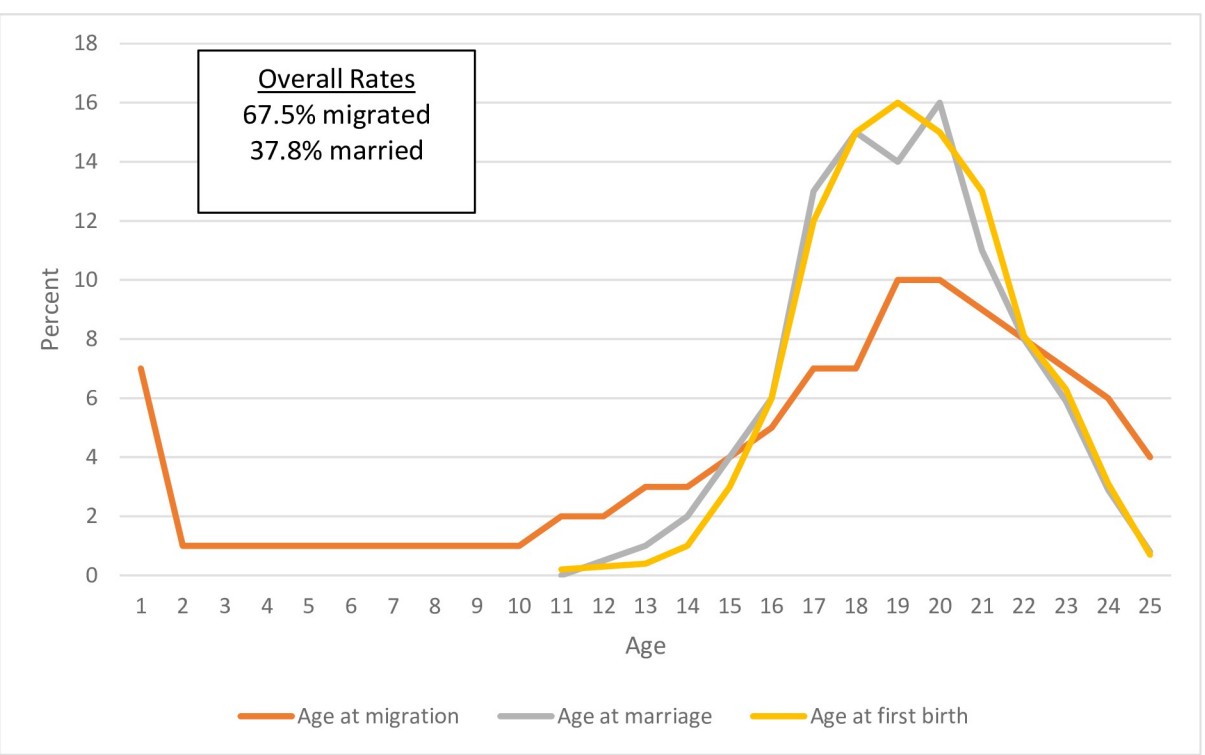

**Fig 1. Age at migration, marriage and first birth.**

Studies, however, differ somewhat in the methods they classify as "modern." In this study, we include standard long-acting reversible (LARC) and short-acting contraception methods, but also emergency contraception, LAM, and the standard days/cycle beads. This classification aligns with the definition of modern contraception of the data we use [62]. Health facility visit was captured through the survey question "In the last 12 months, have you visited a health facility for care for yourself (or your children)?"

Our key independent variable was a recent migration (within the last 1 year) variable that indicated time since migration: never migrated, vs ≤1 years vs >1 years ago (never migrated as the reference). We estimated separate logistic regression models to explore time since migration and its association with contraceptive use and then with recent health facility visit. We also ran these models stratified by ever having given birth since this is strongly related to both outcomes. Models adjust for respondents' current age (in years), educational attainment (four levels: none/ less than primary, primary, secondary/post-primary, tertiary/post-secondary education), tertile of wealth (derived from responses to household asset questions), and country of residence. We also created a variable for when the woman reported wanting a birth (or another birth), categorized as 'years from now', 'soon or months from now', and 'not sure' and adjust for this in exploring modern contraceptive use. All models adjust for clustering of at the village or enumeration area level, as respondents from the same villages are likely more similar than between villages.

Our second set of analyses explores the sequence of events, meaning the order in which women had experienced marriage, birth, and/or migration (for example, she married, then had a birth, then migrated to an urban area). There were sixteen total possible sequences, including for women who had experienced none of these events i.e. never migrated, never married, never had a birth. We created this sequence variable from a woman's age at first

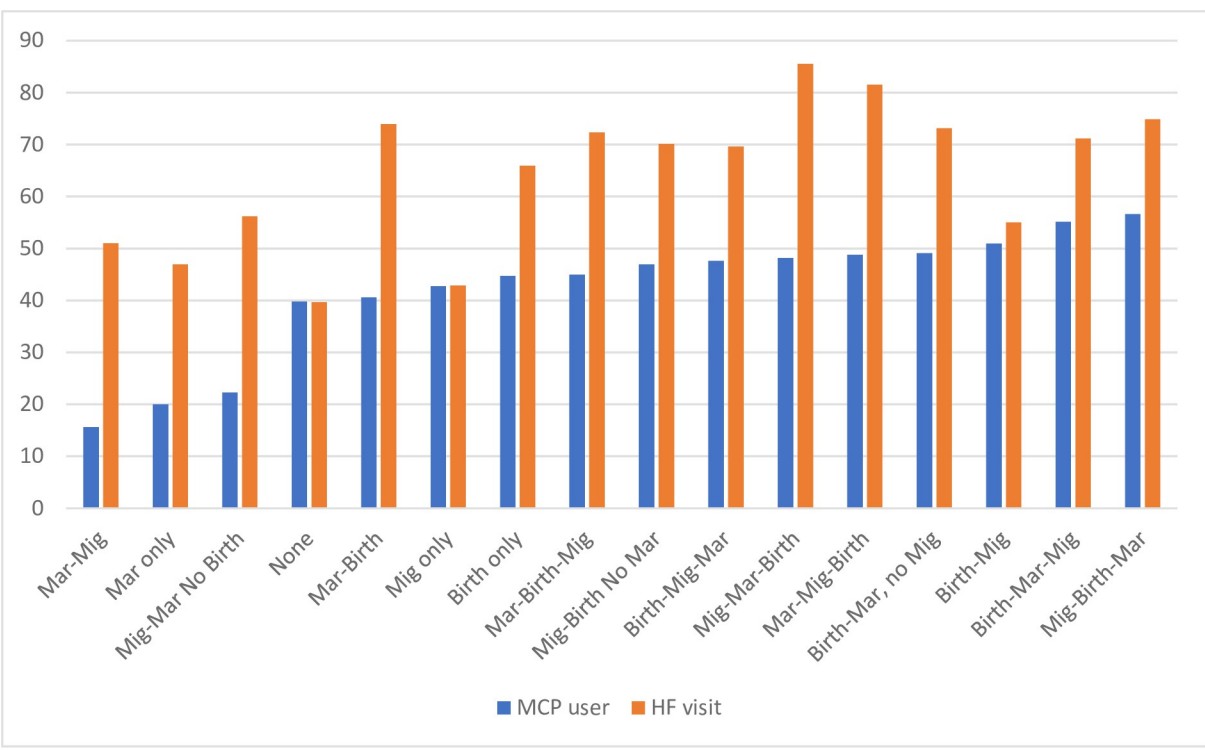

**Fig 2. Proportion of respondents in each category of the sixteen sequence (marriage, migration, birth) combinations using a modern contraceptive method and visiting a health facility in the last 12 months for self or child.**

birth, at marriage, and at migration, calculated from the respondent's date of birth and the year of the event. For age at marriage and age at first birth, values that were less than 10 years were coded as missing as this was likely a data entry error. Migration could occur at any age.

The tabulation of the full set of 16 sequences highlights that the largest group in the dataset are women who have only migrated (29%) followed by women who have neither married, migrated nor had a birth (15%). This distribution is likely due to the young age range of our sample. The sixteen-sequence variable was tabulated against the two key outcomes of interest and illustrated in **Fig 2**, highlighting that women who migrated, then gave birth, then got married are most likely to use a modern method (57%), and those who migrated, then got married, then gave birth were most likely to have visited a health facility in the last 12 months (84%). Scheffe's test for multiple comparisons was used to explore variation in the full sequence variable against key outcomes. These results are shown in the Appendix, **Appendix 1a in S1 Appendix** for modern method use and **Appendix 1b in S1 Appendix** for health facility visit.

We then estimated a series of logistic regression models. The first set of analyses examined the association between recent migration (in the last year) with modern method use and recent health facility visit, including stratified by whether the respondent had ever given birth. The next set of logistic regression models explore the sequence of migration and birth in relation to each outcome.

## Results

Descriptive statistics are shown in **Table 1**, indicate different levels of migration across the six countries, ranging from 35 percent of women who have ever migrated in DRC to 91 percent in Uganda. The sample's education levels are relatively high compared to women in their age

**Table 1. Characteristics by country among urban women 15–24 years of age (who have ever had sex).**

| | DRC | Kenya | Burkina Faso | Nigeria | Uganda | Cote d'Ivoire | Total |
|---|---|---|---|---|---|---|---|
| | N = 1,583 | N = 1,147 | N = 1,456 | N = 401 | N = 676 | N = 962 | N = 6,225 |
| **Demographics** | | | | | | | |
| Age | | | | | | | |
| 15–18 years | 627 (40%) | 272 (24%) | 485 (33%) | 86 (21%) | 219 (32%) | 379 (39%) | 2,068 (33%) |
| 19–24 years | 956 (60%) | 875 (76%) | 971 (67%) | 315 (79%) | 457 (68%) | 583 (61%) | 4,157 (67%) |
| Ever Married | 437 (28%) | 472 (41%) | 684 (47%) | 144 (36%) | 304 (45%) | 309 (32%) | 2,350 (38%) |
| Ever had a birth | 742 (47%) | 620 (54%) | 725 (50%) | 146 (36%) | 377 (56%) | 453 (47%) | 3,063 (49%) |
| Educational Attainment | | | | | | | |
| Primary/Middle school or less | 190 (12%) | 272 (23%) | 542 (38%) | 59 (14%) | 281 (41%) | 426 (45%) | 1,779 (29%) |
| Secondary/post-primary | 1,182 (75%) | 604 (53%) | 819 (56%) | 239 (60%) | 312 (46%) | 452 (47%) | 3,608 (58%) |
| Tertiary/post-secondary | 210 (13%) | 271 (24%) | 95 (7%) | 103 (26%) | 83 (12%) | 84 (9%) | 846 (14%) |
| Wealth tertile | | | | | | | |
| Poorest tertile | 514 (33%) | 140 (12%) | 51 (4%) | 88 (37%) | 110 (16%) | 122 (13%) | 1,025 (17%) |
| Second tertile | 517 (33%) | 471 (41%) | 144 (10%) | 69 (29%) | 174 (26%) | 404 (42%) | 1,779 (29%) |
| Third tertile | 549 (35%) | 536 (47%) | 1,260 (87%) | 81 (34%) | 392 (58%) | 435 (45%) | 3,253 (54%) |
| **Migration** | | | | | | | |
| Never migrated | 1,037 (66%) | 272 (24%) | 398 (27%) | 40 (10%) | 58 (9%) | 218 (23%) | 2,023 (32%) |
| Migrated <1 year ago | 183 (12%) | 333 (29%) | 233 (16%) | 85 (21%) | 209 (31%) | 218 (23%) | 1,261 (20%) |
| Migrated over 1 year ago | 363 (23%) | 542 (47%) | 825 (57%) | 276 (69%) | 409 (61%) | 526 (55%) | 2,941 (47%) |
| **Contraception and health facility visit** | | | | | | | |
| Use MCP | 526 (34%) | 639 (56%) | 668 (47%) | 149 (39%) | 277 (42%) | 387 (41%) | 2,646 (43%) |
| Visit HF in last year | 733 (46%) | 700 (61%) | 1,002 (69%) | 189 (47%) | 436 (65%) | 537 (56%) | 3,597 (58%) |
| Pregnancy intentions | | | | | | | |
| Wait at least a year from now | 1,026 (81%) | 926 (91%) | 1,069 (75%) | 297 (79%) | 277 (85%) | 685 (77%) | 4,280 (81%) |
| Soon/months | 54 (4%) | 79 (8%) | 253 (18%) | 55 (15%) | 39 (12%) | 104 (12%) | 584 (11%) |
| Don't know | 186 (15%) | 8 (1%) | 98 (7%) | 24 (6%) | 10 (3%) | 100 (11%) | 426 (8%) |

group nationally: most women in our sample have some secondary education, ranging from 46 percent in Uganda to 75 percent in DRC. Similarly, wealth levels are quite high with 34 percent (Nigeria) to 87 percent (Burkina Faso) of women in the highest wealth tertile. These higher education and wealth levels are in part due to limiting the sample to women in urban areas. Around 40 percent of women included are currently married, ranging from 28 percent in DRC to 47 percent in Burkina Faso. Almost half of all women have had a birth, ranging from 36 percent in Nigeria to 56 percent in Uganda. Almost all women wanted to have a first pregnancy (or, another pregnancy) at some point. Only 11 percent (ranging from 4 percent in DRC to 18 percent in Burkina Faso) wanted to have a pregnancy soon or within months with most wanting to postpone until years from now.

The analyses that focused on time since migration showed that women who had migrated within the last one year were less likely than urban native women to have used a modern contraceptive method, even adjusting for fertility preferences (OR = 0.76; 95% CI 0.63, 0.91) (**Table 2**). Ever having given birth was positively associated with modern method use, also after adjustment for fertility preferences (OR = 1.58; 95% CI 1.34, 1.86). Women who wanted a birth soon or did not know were less likely to use a modern method, and marriage was not associated with modern method use, except for in the model stratified by ever having given birth; among women who had never given birth for whom being married was associated with lower odds of modern method use (OR = 0.52; 95% CI 0.35, 0.79) likely because married women who have not yet had a birth may be planning or expected to.

**Table 2. Logistic regression of time since migration and factors associated with currently using a modern contraceptive method, and stratified by ever had a birth.**

|  | Model 1: All | Model 2: No birth | Model 3: Ever birth |
|---|---|---|---|
| Ever had a birth | 1.580*** |  |  |
|  | (1.342–1.862) |  |  |
| Migration (REF = never) | REF | REF | REF |
| Migrated within last 1 year | 0.759*** | 0.722** | 0.785* |
|  | (0.632–0.912) | (0.559–0.933) | (0.593–1.038) |
| Migrated over 1 year ago | 1.068 | 0.862 | 1.283** |
|  | (0.918–1.242) | (0.695–1.070) | (1.015–1.621) |
| Age in years | 1.320*** | 1.229** | 1.425*** |
|  | (1.146–1.520) | (1.021–1.479) | (1.137–1.785) |
| Ever married | 0.887 | 0.524*** | 1.049 |
|  | (0.756–1.041) | (0.346–0.793) | (0.856–1.285) |
| Fertility preference (REF = Want pregnancy in years) | REF | REF | REF |
| Want pregnancy soon/months from now | 0.300*** | 0.367*** | 0.303*** |
|  | (0.234–0.384) | (0.250–0.538) | (0.211–0.436) |
| Don't know | 0.856 | 0.768* | 1.077 |
|  | (0.683–1.071) | (0.569–1.037) | (0.727–1.596) |
| Educational Attainment (REF = None, < than Primary) | REF | REF | REF |
| Primary | 1.782*** | 1.496 | 1.568*** |
|  | (1.379–2.303) | (0.900–2.486) | (1.166–2.109) |
| Secondary/post-primary | 2.026*** | 2.165*** | 1.642*** |
|  | (1.572–2.611) | (1.391–3.369) | (1.222–2.207) |
| Tertiary/post-secondary | 2.342*** | 2.720*** | 1.509* |
|  | (1.721–3.188) | (1.668–4.436) | (0.953–2.391) |
| Wealth tertile (REF = Lowest tertile) | REF | REF | REF |
| Middle tertile | 1.190* | 1.193 | 1.212 |
|  | (0.969–1.463) | (0.892–1.595) | (0.876–1.679) |
| Highest tertile | 1.320** | 1.358** | 1.386* |
|  | (1.067–1.632) | (1.015–1.815) | (0.997–1.927) |
| Country (REF = DRC) | REF | REF | REF |
| Kenya | 2.417*** | 1.566*** | 4.048*** |
|  | (1.920–3.043) | (1.144–2.144) | (2.754–5.950) |
| Burkina Faso | 2.229*** | 2.558*** | 1.808*** |
|  | (1.732–2.868) | (1.807–3.621) | (1.290–2.533) |
| Nigeria | 1.457* | 2.959*** | 0.448*** |
|  | (0.964–2.202) | (1.890–4.633) | (0.245–0.818) |
| Uganda | 1.542* | 1.424 | 1.547 |
|  | (0.990–2.404) | (0.759–2.672) | (0.848–2.819) |
| Cote d'Ivoire | 1.957*** | 3.128*** | 1.123 |
|  | (1.407–2.723) | (2.045–4.783) | (0.713–1.768) |

Statistical significance denoted

*p<0.1

** p<0.05

***p<0.01

Among all respondents, time since migration was borderline associated with visiting a health facility in the last 12 months. Ever having given birth was associated with almost three times higher odds of visiting a health facility (OR = 2.92; 95% CI 2.52, 3.37) (**Table 3**). When

**Table 3. Logistic regressions of time since migration and factors associated with recent health facility visit (last 12 months), and stratified by ever had a birth.**

|  | Model 1: All | Model 2: No birth | Model 3: Ever birth |
|---|---|---|---|
| Ever had a birth | 2.915*** |  |  |
|  | (2.521–3.372) |  |  |
| Migration (REF = never) | REF | REF | REF |
| Migrated within last 1 year | 0.835* | 0.998 | 0.677*** |
|  | (0.694–1.007) | (0.786–1.268) | (0.512–0.894) |
| Migrated over 1 year ago | 0.947 | 1.080 | 0.818 |
|  | (0.801–1.119) | (0.883–1.321) | (0.624–1.072) |
| Age in years | 1.038*** | 1.090*** | 0.967* |
|  | (1.011–1.066) | (1.052–1.130) | (0.929–1.006) |
| Ever married | 1.452*** | 1.300** | 1.534*** |
|  | (1.241–1.699) | (1.012–1.670) | (1.270–1.851) |
| Educational Attainment (REF = None, < than Primary) | REF | REF | REF |
| Primary | 1.243* | 1.355 | 1.391** |
|  | (0.961–1.608) | (0.873–2.102) | (1.007–1.921) |
| Secondary/post-primary | 1.632*** | 1.782*** | 1.760*** |
|  | (1.280–2.080) | (1.172–2.709) | (1.297–2.388) |
| Tertiary/post-secondary | 2.045*** | 1.916*** | 2.723*** |
|  | (1.521–2.748) | (1.218–3.013) | (1.693–4.382) |
| Wealth tertile (REF = Lowest tertile) | REF | REF | REF |
| Middle tertile | 1.148 | 1.077 | 1.167 |
|  | (0.931–1.417) | (0.829–1.399) | (0.880–1.548) |
| Highest tertile | 1.395*** | 1.192 | 1.605*** |
|  | (1.108–1.757) | (0.914–1.555) | (1.167–2.207) |
| Country (REF = DRC) | REF | REF | REF |
| Kenya | 1.649*** | 1.743*** | 1.567** |
|  | (1.201–2.262) | (1.260–2.412) | (1.036–2.369) |
| Burkina Faso | 2.564*** | 2.265*** | 3.004*** |
|  | (1.942–3.385) | (1.652–3.104) | (1.979–4.558) |
| Nigeria | 1.194 | 0.759 | 2.137** |
|  | (0.815–1.748) | (0.487–1.185) | (1.099–4.155) |
| Uganda | 2.189*** | 1.740** | 2.835*** |
|  | (1.453–3.298) | (1.120–2.702) | (1.549–5.188) |
| Cote d'Ivoire | 1.818*** | 1.323 | 2.814*** |
|  | (1.288–2.566) | (0.886–1.976) | (1.641–4.826) |

Statistical significance denoted

*p<0.1

** p<0.05

***p<0.01

stratified by ever having had a birth, women who had ever had a birth were less likely to have a recent health facility visit if they were recent migrants (OR = 0.68; 95% CI 0.51, 0.89). Women who were married were much more likely to visit a health facility. Educational attainment was highly associated with visiting a health facility, and being in the highest wealth tertile was highly associated for women who had had a birth though not for those who had never had a birth.

Exploring the sequence of events highlighted that while migration alone was not associated with the outcomes, the combination of having given birth and having migrated were the important combination regardless of the order. About half of the young women had migrated

or given birth, with 34 percent experiencing both and 17 percent neither. The logistic regression model highlights the combination associated with modern method use. Compared to women who had migrated or given birth, women who had experienced both had higher odds of modern method use (OR = 1.40; 95% CI 1.19, 1.64) (**Table 4; Model 1).** The odds of modern method use increased with educational attainment and in the highest wealth category, and was significantly lower if women wanted a birth soon compared to years from now.

For recent health facility, women who had neither given birth or migrated were less likely to have visited (OR = 0.72; 95% CI 0.61, 0.85) and women who had both given birth and migrated were more likely to (OR = 1.75; 95% CI 1.48, 2.06) compared to women who had either migrated or given birth but not both (**Table 4**; **Model 2**). Women who were married were much more likely to have visited a health facility recently (OR = 1.84; 95% CI 1.58, 2.14). Higher levels of education were associated with visiting a health facility in the last 12 months, and highest wealth tertile was borderline significant.

## Discussion

This article examines internal migration and young women's use of contraception in Burkina Faso, Côte d'Ivoire, Democratic Republic of Congo, Kenya, Nigeria, and Uganda. Our data show high overlap in the ages at which girls are migrating, getting married and having their first birth, with migration peaking at around 18 years. The common confluence of these events raises questions about how they are associated with contraceptive use during the transition to adulthood, and how their sequence as well as the recency of migration may affect outcomes related to accessing health services and modern contraception in particular.

We find some support for the disruption hypothesis in that women who have migrated within the last year are less likely to use a modern method, even after adjusting for fertility desires. This might be because they have not yet integrated into a healthcare network in the early stages of arriving in a new place. Potentially they cannot afford or access contraception, although it is possible if they migrated without their partner may not have a need for contraception. The results also show that preferences around timing of pregnancy influence the use of modern contraception: women who want a pregnancy soon are less likely to use a modern method. Capturing true fertility intentions is challenging, as women may experience ambivalence or even contradictory feelings regarding a potential pregnancy [63]. Ambivalent responses are common in surveys, for example where women report wanting to delay pregnancy but also report that it would be no problem or only a small problem to get pregnant soon [64]. Amongst women who have never had a birth, women who are married are less likely to use a modern contraceptive method, which may reflect the desires and expectations for pregnancy following marriage from the woman, her spouse, and larger society. Given these associations, it is critical for policymakers and health providers to consider these desires in programming.

Women who migrated within the last year were less likely to have recently visited a health facility, particularly if they had ever given birth. For recent visit to a health facility, the most salient factor was whether women had given birth. In part, this finding is to be expected given that the variable captures whether women visited the facility for their own health or for their child's health, but it also likely reflects how pregnancy and birth integrate women into healthcare systems through pre and postnatal care. In some settings, women may migrate alone or with a spouse leaving their children behind. Potentially, this is the reason some women who have had a birth had not visited a health facility. However, if women did migrate with their children, future programs should ensure their immediate access to health services. The WHO recommends every pregnant women and newborn receives quality care throughout pregnancy, childbirth and the postnatal period [65, 66].

**Table 4. Factors associated with modern contraceptive use (Model 1) and recent health facility visit (Model 2) including sequence of migration and birth.**

| | Model 1: Modern Method use | Model 2: Recent facility visit |
|---|---|---|
| Sequence (REF = Migration or Birth) | REF | REF |
| Neither migration or birth | 1.033 | 0.718*** |
| | (0.863–1.236) | (0.608–0.847) |
| Both migration AND birth | 1.400*** | 1.749*** |
| | (1.194–1.642) | (1.483–2.064) |
| Age in female respondent questionnaire | 1.067*** | 1.066*** |
| | (1.037–1.097) | (1.039–1.094) |
| Ever married | 0.989 | 1.836*** |
| | (0.849–1.151) | (1.575–2.141) |
| Fertility preference (REF = Want pregnancy in years) | REF | REF |
| Want pregnancy soon/months from now | 0.275*** | - |
| | (0.215–0.353) | - |
| Don't know | 0.826 | - |
| | (0.656–1.039) | - |
| Educational Attainment (REF = None, < than Primary) | REF | REF |
| Primary | 1.788*** | 1.264* |
| | (1.384–2.311) | (0.972–1.644) |
| Secondary/post-primary | 2.005*** | 1.556*** |
| | (1.560–2.576) | (1.218–1.989) |
| Tertiary/post-secondary | 2.145*** | 1.704*** |
| | (1.585–2.901) | (1.270–2.288) |
| Wealth tertile (REF = Lowest tertile) | REF | REF |
| Middle tertile | 1.140 | 1.094 |
| | (0.931–1.398) | (0.890–1.345) |
| Highest tertile | 1.247** | 1.247* |
| | (1.010–1.539) | (0.998–1.560) |
| Country (REF = DRC) | REF | REF |
| Kenya | 2.215*** | 1.327* |
| | (1.764–2.782) | (0.968–1.818) |
| Burkina Faso | 2.115*** | 2.022*** |
| | (1.643–2.723) | (1.543–2.650) |
| Nigeria | 1.317 | 0.816 |
| | (0.875–1.983) | (0.561–1.187) |
| Uganda | 1.390 | 1.685*** |
| | (0.897–2.154) | (1.133–2.506) |
| Cote d'Ivoire | 1.868*** | 1.499** |
| | (1.349–2.587) | (1.078–2.087) |

Statistical significance denoted

*p<0.1

** p<0.05

***p<0.01

The sequence of events were not very significant overall, however, experience of both migration and a birth were the most likely to be current modern contraceptive users and to have visited a health facility recently. While the recent migration may suggest an initial disruption, these findings suggest that migrant women, particularly who have given birth, have improved uptake of modern contraceptive methods (after adjusting for fertility preferences) and higher odds of visiting a health facility. Education and wealth were significantly associated as well. Overall these results may suggest improved outcomes for this group.

This study has several limitations. First, the datasets are cross-sectional, with one survey per country, so it was not possible to compare before and after migration. It was also not possible to extract if the migration was rural to urban, or urban to urban. If urban women generally have a health advantage, then better outcomes may be expected for urban-to-urban women, potentially weakening observed associations between migration and outcomes if they are worse for rural to urban and better for urban to urban. It was also not possible to know who the respondent migrated with, for example, some women may have migrated leaving young children at home with family, or without their spouse. Lastly, it was not possible to measure the reason for migration, which would be useful to understand if women migrated for marriage, for work, or other reasons that may relate to her social and economic standing after migration and also her pregnancy risks.

We found significant variation by country, which may in part reflect differences in data collection or sampling, but likely also reflects underlying unique patterns. For example, countries with more economic and education opportunities, with better infrastructure, see more migration to urban areas [67]. Gendered cultural norms and types of employment may lead to gender differentiation in patterns of migration observed [67]. While urban settings generally have an advantage in terms of access to services and resources [46], there are neighborhoods and sub-groups which may fail to access these. Recent migrant arrivals may require additional support to link with health services and access key needs such as modern contraception.

Overall, our findings highlight variation in modern contraceptive use and recent health facility visit for urban young women in six African countries. Having a birth was the most important factor, more so than migration or marriage, in using a modern contraceptive method and recent health facility visit, suggesting that a birth may link women into formal health services and that contraception may be used in this young age range for birth spacing. Migrating within the last year was associated with lower odds of modern method use, suggesting a potential disruption due to migration and high-risk period for pregnancy. Programs and policies should support young migrant women upon arrival to ensure they have access to contraception and other health needs. Young migrant mothers may be less likely to have visited a health facility recently, although it is unclear if this is a gap or may reflect the mother's desire for another pregnancy soon. Understanding how fertility preferences, migration and having children intersect in this young age group is important to ensure the health of women and their families during a transitional period with potential disruptions in access to services and information.

## Supporting information

**S1 File. Inclusivity questionnaire.**
(DOCX)

**S1 Appendix.**
(DOCX)

## Author Contributions

**Conceptualization:** Jessie Pinchoff, Isabel Pike, Karen Austrian, Kathryn Grace, Caroline Kabiru.

**Data curation:** Jessie Pinchoff.

**Formal analysis:** Jessie Pinchoff.

**Investigation:** Jessie Pinchoff, Isabel Pike, Kathryn Grace, Caroline Kabiru.

**Methodology:** Jessie Pinchoff, Kathryn Grace.

**Resources:** Karen Austrian.

**Supervision:** Karen Austrian.

**Writing – original draft:** Jessie Pinchoff, Isabel Pike.

**Writing – review & editing:** Karen Austrian, Kathryn Grace, Caroline Kabiru.

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
