## [Decision Letter · Decision Letter 0]

30 Apr 2024

PONE-D-24-07930How Migration Shapes Modern Contraceptive Use among Urban Young Women: Evidence from Six African countriesPLOS ONE

Dear Dr. Pinchoff,

Thank you for submitting your manuscript to PLOS ONE. After careful consideration, we feel that it has merit but does not fully meet PLOS ONE’s publication criteria as it currently stands. Therefore, we invite you to submit a revised version of the manuscript that addresses the points raised during the review process.

We look forward to receiving your revised manuscript.

Kind regards,

Patrick Ifeanyi Okonta, MBBCh, MPH, FWACS, FMCOG, MD, DRH

Academic Editor

PLOS ONE

Journal Requirements:

Reviewers' comments:

Reviewer's Responses to Questions

**Comments to the Author**

1. Is the manuscript technically sound, and do the data support the conclusions?

Reviewer #1: Yes

Reviewer #2: Yes

2. Has the statistical analysis been performed appropriately and rigorously? 

Reviewer #1: No

Reviewer #2: Yes

3. Have the authors made all data underlying the findings in their manuscript fully available?

Reviewer #1: Yes

Reviewer #2: Yes

4. Is the manuscript presented in an intelligible fashion and written in standard English?

Reviewer #1: Yes

Reviewer #2: Yes

5. Review Comments to the Author

Reviewer #1: Title

The authors have addressed an important area of population health that link migration to facility visits and contraceptive among use adolescent girls and young women. I have generally enjoyed reading the manuscript but there are issues that must be addressed before the paper may be considered for publication.

Abstract:

1.There is minor grammatical error in the abstract line 31

2. May the authors include key results from their logistic regression model (Odds or Odds ratios and their corresponding 95% confidence intervals) in their result section of their abstract.

Background

The authors have presented important information in the background. However, they have not indicated the proportion of migrants that are young aged 15 to 24. Demographic and Health Surveys have information on these estimates for both. The authors alluded to the scarcity of studies on migration and family planning use, I agree with the authors but there are studies they could have referenced including their quantitative findings e.g. contraceptive prevalence rate among young women, odd ratio of contraceptive use comparing migrants to non-migrants.

Methods

This study used data collected in 6 African countries. Observations within countries could be correlated; this should have been investigated to assess evidence or significant correlation. The authors needed to apply multilevel modeling approach or models that can adjust for clustering of observations within countries other than using naïve logistic regression model. The clustering could even apply at countries level if there were clusters within countries. Use of naïve logistic regression violates the independence of observations assumption of the generalized linear models to which logistic is part of. These need to be addressed.

Discussion

The discussion is well written. The authors mentioned in the discussion that there were wide variations in the exposure (migration) and outcomes, this is consistent with clustering as suggested in the methods section. The authors will need to rework results based on my suggestion in methods and update both results and methods.

Reviewer #2: This is a nice paper and my comments are mostly pretty minor. I thought it was very well written- the intro and framing in particular. The research question itself, measured outcomes, and reason for questions are all clear. Its kind of hard to draw any highly relevant policy conclusions from the somewhat mixed findings, but that is research.

My comments are below. First two seem critical to me, others are more minor.

1) Given that this is a life-course analysis, why did you use cross sectional rather than longitudinal data that is available in PMA? Were the migration questions not asked often enough? Or did you not have a large enough sample with life events in the longitudinal data? Please address why you made that analytic choice, because the obvious choice would be to look at migrants in the panel data, so you can follow individual women over time. (I am sure there was a good reason, its just good to explain the choice to the reader)

2) On pg 6, line 205- you say that you are using 2 yrs as a cutoff point for recent migration, but on page 7 line 249 you say migration within the last year is recent migration. Again on line 286, you say within the last yr is recent migration, but in 307-308, you define based on a 2 yr cutoff. Looks like the actual regressions use one year- but text switches back constantly between describing 1 yr and 2 yrs as ‘recent migration’. I’m guessing this is because there were several versions of the analysis done and you landed on the 1 yr timeframe- and you just need to update the text. But if there is a substantive reason that you use the two different time periods to define ‘recent’ and you used them both in different parts of the analysis somehow, then please make that clearer to the reader.

3) The citation style switches back and forth constantly throughout the text (sometimes it’s a number, sometimes its author and yr). Presumably from multiple authors and not a big deal, but definitely needs fixing.

4) Also in the category of ‘update text’ the education categories described in lines 313 314 should be changed to match the regression wording (which makes more sense, actually- when I first read the text I thought you had overlapping categories with “secondary” in two different categories)

5) I would like a sentence or two more explanation of how order for first birth-migration might matter (lines 254- 257). You just say ‘in theory it might matter if women who gave birth before migrating are less likely to have visited a facility”.. But alternately, I might argue women who migrate rural-urban may have *more* access to facilities than they did in rural area, where health care infrastructure can be minimal. Since this 'order' question is a key part of the analysis, i think you should spend a few more sentences explaining.

6) What are the scheffe’s tables actually adding? If they are important to the analysis, say something about them in the text. If not, I’d pop them in an appendix.

7) Are you really certain 91% of 15-24 yr olds in urban Uganda have migrated? If that is what the data says- then okay. But it seems implausibly high.

6. PLOS authors have the option to publish the peer review history of their article (what does this mean?). If published, this will include your full peer review and any attached files.

Reviewer #1: **Yes: **Reuben Christopher Moyo

Reviewer #2: No

---

## [Author Response · Author response to Decision Letter 0]

24 Jun 2024

This has been completed and uploaded as Supporting Information.

We have a complete Data Availability statement, but all of our data is publicly available and this is a secondary data analysis.

We have added a sentence at line 273 that we did not require IRB approval because this is entirely a secondary data analysis of publicly available data.

Reviewers' comments:

Reviewer's Responses to Questions

Comments to the Author

5. Review Comments to the Author

Reviewer #1: Title

The authors have addressed an important area of population health that link migration to facility visits and contraceptive among use adolescent girls and young women. I have generally enjoyed reading the manuscript but there are issues that must be addressed before the paper may be considered for publication.

Abstract:

1.There is minor grammatical error in the abstract line 31

We have corrected this error, thank you for catching it.

2. May the authors include key results from their logistic regression model (Odds or Odds ratios and their corresponding 95% confidence intervals) in their result section of their abstract.

We have added key results with numbers in the results section of the abstract.

Background

The authors have presented important information in the background. However, they have not indicated the proportion of migrants that are young aged 15 to 24. Demographic and Health Surveys have information on these estimates for both. The authors alluded to the scarcity of studies on migration and family planning use, I agree with the authors but there are studies they could have referenced including their quantitative findings e.g. contraceptive prevalence rate among young women, odd ratio of contraceptive use comparing migrants to non-migrants.

We have added a statistic in the methods section, and have some discussion of migration and gender literature, however most statistics available refer to international migration, often from parts of Africa to Europe or the Middle East, and very few were found that relate to more the rural-urban within-country movement. We found it extremely challenging to find estimates of this kind of urban mobility within a country.

We also do have our own statistics from the PMA but have left them in the results section. We did not quantify the relationship between migration and each outcome but rather the more complex versions related to migration in the sequence of events and recency of the migration (so we do not present migrants vs non-migrants). 

Methods

This study used data collected in 6 African countries. Observations within countries could be correlated; this should have been investigated to assess evidence or significant correlation. The authors needed to apply multilevel modeling approach or models that can adjust for clustering of observations within countries other than using naïve logistic regression model. The clustering could even apply at countries level if there were clusters within countries. Use of naïve logistic regression violates the independence of observations assumption of the generalized linear models to which logistic is part of. These need to be addressed.

This was a major oversight in the writing of the methods section! We have added a sentence at line 320. All models adjust for the enumeration area or village level to adjust the standard errors and account for the clustering of observations within villages, and we also adjust for the country. This was done in all models presented, we just did not write this information in the methods!

Discussion

The discussion is well written. The authors mentioned in the discussion that there were wide variations in the exposure (migration) and outcomes, this is consistent with clustering as suggested in the methods section. The authors will need to rework results based on my suggestion in methods and update both results and methods.

Thank you! Yes the results and tables reflect models correctly adjusted for clustering at the enumeration area or cluster/village level. We have added this critical detail to the methods.

Reviewer #2: This is a nice paper and my comments are mostly pretty minor. I thought it was very well written- the intro and framing in particular. The research question itself, measured outcomes, and reason for questions are all clear. Its kind of hard to draw any highly relevant policy conclusions from the somewhat mixed findings, but that is research.

Thank you, we agree the findings are a bit mixed, but the potential disruptive effect of migration for such young women arriving in urban areas may highlight the need to tailor interventions or services to meet their needs.

My comments are below. First two seem critical to me, others are more minor.

1) Given that this is a life-course analysis, why did you use cross sectional rather than longitudinal data that is available in PMA? Were the migration questions not asked often enough? Or did you not have a large enough sample with life events in the longitudinal data? Please address why you made that analytic choice, because the obvious choice would be to look at migrants in the panel data, so you can follow individual women over time. (I am sure there was a good reason, its just good to explain the choice to the reader)

This is for a few reasons, first, yes the sample size was not very large with the longitudinal data, as only a subset of women were followed. The survey also was not designed to capture information about migration, so it wont capture migration very well as women who migrated may have been less likely to be followed up. We worried about attrition in the follow up surveys, and differential loss to follow up, as again these surveys are not aiming to capture migration.

We decided to use the cross-sectional survey data for all women available using their most recent information and capture historical migration. Future analyses that track women in their movement within a country will be very useful, but the study must be carefully designed for this purpose.

2) On pg 6, line 205- you say that you are using 2 yrs as a cutoff point for recent migration, but on page 7 line 249 you say migration within the last year is recent migration. Again on line 286, you say within the last yr is recent migration, but in 307-308, you define based on a 2 yr cutoff. Looks like the actual regressions use one year- but text switches back constantly between describing 1 yr and 2 yrs as ‘recent migration’. I’m guessing this is because there were several versions of the analysis done and you landed on the 1 yr timeframe- and you just need to update the text. But if there is a substantive reason that you use the two different time periods to define ‘recent’ and you used them both in different parts of the analysis somehow, then please make that clearer to the reader.

Yes this is correct, we ran the analysis both ways and had very similar overall findings, choosing to present the 1 year version here. We have fixed the issues throughout and aligned.

3) The citation style switches back and forth constantly throughout the text (sometimes it’s a number, sometimes its author and yr). Presumably from multiple authors and not a big deal, but definitely needs fixing.

Thank you for catching this ! We have aligned all references to be numeric.

4) Also in the category of ‘update text’ the education categories described in lines 313 314 should be changed to match the regression wording (which makes more sense, actually- when I first read the text I thought you had overlapping categories with “secondary” in two different categories)

Agreed, these categories are a bit confusing but we have aligned the categories based on how the variable is derived and to ensure no overlap in the titles. They basically mean everything up to completing secondary vs actually completed secondary, but it was not clear, we hope now it is. Thank you!

5) I would like a sentence or two more explanation of how order for first birth-migration might matter (lines 254- 257). You just say ‘in theory it might matter if women who gave birth before migrating are less likely to have visited a facility”.. But alternately, I might argue women who migrate rural-urban may have *more* access to facilities than they did in rural area, where health care infrastructure can be minimal. Since this 'order' question is a key part of the analysis, i think you should spend a few more sentences explaining.

We have added some additional thoughts here and a new reference on the subject.

6) What are the scheffe’s tables actually adding? If they are important to the analysis, say something about them in the text. If not, I’d pop them in an appendix.

We debated this and agree we will move them into an appendix. They do not add so much on their own.

7) Are you really certain 91% of 15-24 yr olds in urban Uganda have migrated? If that is what the data says- then okay. But it seems implausibly high.

91% is quite high, we went back and confirmed. The proportion is even quite high among the full dataset of 15-24 year olds (85%) before we make restrictions. Potentially this is related to how they surveyed in Uganda.

---

## [Editor Report · Decision Letter 1]

2 Jul 2024

How Migration Shapes Modern Contraceptive Use among Urban Young Women: Evidence from Six African countries

PONE-D-24-07930R1

Dear Dr. Pinchoff,

We’re pleased to inform you that your manuscript has been judged scientifically suitable for publication and will be formally accepted for publication once it meets all outstanding technical requirements.

Kind regards,

Patrick Ifeanyi Okonta, MBBCh, MPH, FWACS, FMCOG, MD, DRH

Academic Editor

PLOS ONE
---

## [Editor Report · Acceptance letter]

12 Jul 2024

PONE-D-24-07930R1 

PLOS ONE

Dear Dr. Pinchoff, 

I'm pleased to inform you that your manuscript has been deemed suitable for publication in PLOS ONE. Congratulations! Your manuscript is now being handed over to our production team.

Kind regards, 

on behalf of

Professor Patrick Ifeanyi Okonta 

Academic Editor

PLOS ONE